# COVID-19 vaccine hesitancy among the adult population in Bangladesh: A nationwide cross-sectional survey

**Mohammad Bellal Hossain**[1]*, **Md. Zakiul Alam**[1], **Md. Syful Islam**[2], **Shafayat Sultan**[1], **Md. Mahir Faysal**[1], **Sharmin Rima**[3], **Md. Anwer Hossain**[1], **Abdullah Al Mamun**[1]

1 Department of Population Sciences, University of Dhaka, Dhaka, Bangladesh, 2 Department of Population Science, Jatiya Kabi Kazi Nazrul Islam University, Mymensingh, Bangladesh, 3 Ovibashi Karmi Unnayan Program (OKUP), Dhaka, Bangladesh

* bellal@du.ac.bd

**Data Availability Statement:** Data are available at: http://dx.doi.org/10.17632/prgh4bb3yf.1.

**Funding:** The author(s) received no specific funding for this work.

## Abstract

### Introduction

Studies related to the COVID-19 vaccine hesitancy are scanty in Bangladesh, despite the growing necessity of understanding the population behavior related to vaccination. Thus, the present study was conducted to assess the prevalence of the COVID-19 vaccine hesitancy and its associated factors in Bangladesh to fill the knowledge gap.

### Methods and materials

This study adopted a cross-sectional design to collect data from 1497 respondents using online (Google forms) and face-to-face interviews from eight administrative divisions of Bangladesh between 1–7 February 2021. We employed descriptive statistics and multiple logistic regression analysis.

### Results

The prevalence of vaccine hesitancy was 46.2%. The Muslims (aOR = 1.80, p ≤ 0.01) and the respondents living in the city corporation areas (aOR = 2.14, p ≤0.001) had more hesitancy. There was significant variation in vaccine hesitancy by administrative divisions (geographic regions). Compared to the Sylhet division, the participants from Khulna (aOR = 1.31, p ≤0.001) had higher hesitancy. The vaccine hesitancy tended to decrease with increasing knowledge about the vaccine (aOR = 0.88, p≤0.001) and the vaccination process (aOR = 0.91, p ≤ 0.01). On the other hand, hesitancy increased with the increased negative attitudes towards the vaccine (aOR = 1.17, p≤0.001) and conspiracy beliefs towards the COVID-19 vaccine (aOR = 1.04, p≤0.01). The perceived benefits of COVID-19 vaccination (aOR = 0.85, p≤0.001) were negatively associated with hesitancy, while perceived barriers (aOR = 1.16, p ≤0.001) were positively associated. The participants were more hesitant to accept the vaccine from a specific country of origin (India, USA, Europe).

**Competing interests:** The authors have declared that no competing interests exist.

## Conclusions

Our findings warrant that a vigorous behavior change communication campaign should be designed and implemented to demystify negative public attitudes and conspiracy beliefs regarding the COVID-19 Vaccine in Bangladesh. The policymakers should also think about revisiting the policy of the online registration process to receive the COVID-19 vaccine, as online registration is a key structural barrier for many due to the persistent digital divide in the country. Finally, the government should consider the population's preference regarding vaccines' country of manufacture to reduce the COVID-19 vaccine hesitancy.

## Introduction

The development of vaccination is significant public health-related progress though anti-vaccination attitudes, behavior, and associated misconceptions are widely prevalent [1]. The evidence shows that the effectiveness of vaccination programs has been affected by *vaccine hesitancy* [2], where hesitancy has been defined as a "delay in acceptance or refusal of vaccination despite the availability of vaccination services" [3]. Hesitancy regarding the Coronavirus diseases 19 (COVID-19) vaccination is prominently visible around the world [4] in such a period when the effort towards reaching herd immunity has been targeted to achieve through the vaccination coverage [5].

The Government of Bangladesh (GoB) has launched the biggest-ever vaccination program nationwide to vaccinate 80% (over 130 million) of the country's total population with the COVID-19 vaccines in four stages [6] though nearly 34% population are below 18 years old [7]. The GoB has published a national deployment and vaccination plan for the COVID-19 vaccination that requires an online registration to receive the COVID-19 vaccine. According to the plan, the GoB has targeted to vaccinate 117 million population aged 18 years and above. However, as of 3 August 2021, only 14% of the targeted people have been registered to receive vaccination, of whom 57% received the 1st dose vaccine with a significant variation by gender (men 61%, women 39%) and administrative regions (Dhaka (19%) received more vaccination than other administrative divisions) [8]. Overall, only 8% of the targeted population have received the first dose of vaccination, while 4% received the second dose of vaccination [8]. Simultaneously, incidents about the lack of interest among the population about the vaccine uptake and lack of response about the registration process have been repeatedly reported in the media, showing vaccine hesitancy among the population.

The studies conducted around the world to explore the COVID-19 vaccine hesitancy has shown that various socio-economic and demographic variables, different constructs of health belief model (HBM) [9, 10], level of knowledge related to the vaccine [11, 12], attitude towards COVID-19 vaccination [12, 13], conspiracy beliefs regarding the origin, effectiveness, and consequences of receiving vaccines [14, 15], preventive behavioral practices related to COVID-19 [16, 17], newness, safety, and probable side effects of the vaccine [18] as primarily responsible for vaccine hesitancy.

Studies related to vaccine hesitancy, COVID-19, or other diseases, are scanty in Bangladesh, despite the growing necessity of understanding the people's vaccination-related behavior. Few studies have been conducted in Bangladesh to assess the COVID-19 vaccine hesitancy, which has reported a vaccine hesitancy rate between 25.4% to 50% [13, 19–22]. However, the findings of these studies are not representative of the context of Bangladesh and have the following

limitations: small sample size [20, 21]; collecting data using only the online platform [19–21] despite having an apparent existence of a digital divide across the country [23]; and non-use of psychological and behavioral variables related to vaccine hesitancy [22]. Thus, a nationwide survey was conducted to assess the prevalence of the COVID-19 vaccine hesitancy and its associated factors in Bangladesh to fill the knowledge gap recruiting respondents from all the eight administrative divisions of the country.

## Materials and methods

### Description of the study setting

This study was conducted in Bangladesh. The country has a strong primary healthcare system that provides services at the doorsteps of citizens. As a result, it has a successful childhood vaccination coverage of more than 85% [24]. Bangladesh has also made remarkable progress in poverty reduction, maintained by sustained economic growth. Poverty has declined from 43.5% in 1991 to 14.3% in 2016, based on the international poverty line of $1.90 per day [25]. The country has a 166.5 million population, which is about 2.11% of the world population. However, most of the population is still very young, with a median age of 27.6 years. The rate of urban population is 37.4% of the total population. The population density is 1125 per square kilometer, and 88.4% of the population are Muslims. The sex ratio is 100.2, and the total fertility rate is 2.04 [26]. The life expectancy at birth is 72.6 years (71.1 for men and 74.2 for women). The literacy rate is 74.7% among the adult population aged 15 years and above (77.4% for men and 71.9% for women) [26]. We collected data from all the eight administrative divisions of Bangladesh. S1 Map shows the districts from where data for this study were collected. The samples were proportionately distributed to the population size of the divisions.

### Study population and inclusion and exclusion criteria

This study was conducted among the population aged 18 years and above using a cross-sectional research design. Thus, the population aged 18 years and above, living in Bangladesh, and knowing about the COVID-19 vaccine were used as the selection criteria for the face-to-face interview. The age of 18 years was considered because the vaccine was not available for people younger than 18 years when this study was conducted. In addition to the criteria used in the face-to-face interview, reading and writing and internet use were used as the selection criteria for the online survey. On the other hand, pregnancy, breastfeeding, and the presence of any severe chronic illness were considered as the exclusion criteria for selecting respondents for this study.

### Sample size

We used the following formula to calculate the sample size: $(Z^2pq/e^2)Deff^*NR$. We used Z-score for 95% confidence interval (Z = 1.96), prevalence (p) of willingness to accept a COVID-19 vaccine from an earlier study (p = 0.325) [13], margin of error (e = 0.03); design effect (Deff = 1.6) for sampling variation; and a non-response rate (NR = 10%). The calculated sample size was 1635, distributed for face-to-face and online surveys using a 2:1 ratio considering the country's digital divide. However, 112 respondents did not consent to participate in the study (101 in the face-to-face survey and 11 in the online survey). The response rate was 93.1 percent (91.9% in face-to-face surveys and 97.7% in online surveys). We also had to exclude 26 respondents who did not know about the COVID-19 vaccine. Thus, the final sample for this study was 1497 for analysis (1022 (68.3%) from face-to-face survey and 475 (31.7%) from online survey). The data is now placed in the Mendeley open research data repository [27].

## Modes of data collection

Data were collected between 1–7 February 2021. We collected data using both online and face-to-face interviews. The online data were collected through Google Forms using the Bengali language. The participants to whom the survey link was sent through e-mail, WhatsApp, or Facebook were requested to fill-up the form and circulate the link in their network to reach more people. In addition, the research team members circulated the survey link in their respective professional and social networks through the snowball process. The online link was valid for three days. The online data were downloaded, and divisional distribution was assessed. Data were then collected from the remaining sample size for each division through a face-to-face interview using quota sampling. We used a non-probability sampling technique as the complete list of the adult population was not available. Due to budgetary constraints, a listing of the households was also not possible. The duration for the face-to-face data collection was four days. The graduate and post-graduate level students of the University of Dhaka were recruited and trained to collect the data. We trained the data collectors through the online platform google meet. The training included discussions on how to conduct face-to-face data collection and quota sampling strategy.

## Ethical approval and participant's consent

We took ethical approval (registration number - 391310l2021) from the National Research Ethics Committee of the Bangladesh Medical Research Council (BMRC). Participation in this study was entirely voluntary, and no incentive was provided to the participants. For the face-to-face interview, the interviewer informed the scopes and implications of the study to the respondents and requested to participate voluntarily. The interviewer did not interview the respondents if they declined to participate. For the online survey, the respondents voluntary and informed consent was sought by using the question "do you agree to participate in this study after reading the information about this research?" which had a binary response option. The respondents who consented to participate voluntarily in the survey then needed to click on the "Continue" option and only then were they directed to complete the Google Forms. The respondents could not participate in this study if their answers to this consent question were "no."

## Measures

We selected variables and items for constructing scales from the previous studies [12, 28–32] and then mixed and customized the different items to develop the scale for the Bangladesh context. The data collection tool was pretested to validate using the face-to-face interview to determine respondents' understanding of the questions, comprehensiveness of the questionnaire, and wording, length, and sensitivity of the questions. We calculated internal consistency using Cronbach's alpha ($\alpha$) to assess the reliability of the items used in the scales. We developed a total of ten scales, of which seven scales had an $\alpha$ between 0.700 to 0.857, and three had an $\alpha$ between 0.612 to 0.657. The reliability analysis of seventy percent of our scales was good as the $\alpha$ ranged between 0.7 and 0.8 [33]. The discussion below provides a detailed discussion on scale development, and the items used in these scales are available at https://osf.io/e4xph/.

**Outcome variable: Vaccine hesitancy.** We used two questions to measure this study's outcome variable, which is the COVID-19 vaccine hesitancy. First, we asked the respondents what they would do if they got the chance to take the COVID 19 vaccine for free? The responses to this question were: 1 = Surely, I will take it; 2 = Probably I will take it; 3 = I will delay taking it; 4 = I am not sure what I will do; 5 = Probably I will not take it; 6 = Surely, I will not take it. The responses 1 = Surely, I will take it and 2 = Probably I will take it were

considered vaccine acceptancy, and the rest indicated vaccine hesitancy. The second question was what they would do if their family or friends thought of taking COVID 19 vaccine? The responses to this question were: 1 = Strongly encourage them; 2 = Encourage them; 3 = Ask them to delay getting the vaccine; 4 = I will not say anything about it; 5 = Discourage them to take the vaccine; 6 = Forbid them to take the vaccine. The responses 1 = Strongly encourage them and 2 = Encourage them were considered vaccine acceptancy, while the rest indicated vaccine hesitancy. The Cronbach Alpha (α) of these two items was 0.833, which shows good internal consistency. We combined these two items and calculated the prevalence of vaccine hesitancy if the respondents had hesitancy in any of the two items.

**Independent variables.**  *Socio-economic and demographic variables*. We included the following socio-economic and demographic variables as the independent variables of this study: age, sex, religion, marital status, educational attainment, place of residence, administrative division, occupation, number of household members, and household income.

*Behavioral practice to prevent COVID-19*. We measured preventive behavioral practices related to COVID-19 using three four-point Likert scale items. The total score of these items ranged between 3 and 12, with a higher score indicating better preventive practices with the Cronbach alpha (α) 0.857.

*Knowledge about the COVID-19 vaccine*. We assessed the knowledge related to the COVID-19 vaccine using four five-point Likert scale questions. The total score of these items ranged between 4 and 20, with a higher score indicating higher knowledge with the Cronbach alpha (α) 0.643.

*Knowledge about the vaccination process*. Knowledge about the COVID-19 vaccination process was measured by six binary (yes = 1, no = 0) questions. The Cronbach alpha (α) of these six questions was 0.765, which shows good internal consistency. Thus, the higher scores indicated better knowledge.

*COVID-19 vaccine conspiracy*. Conspiracy related to the COVID-19 vaccine was measured using nine five-point Likert scale items (α = 0.716). The total score of these items ranged between 9 and 45, where a higher score indicated having higher conspiracy beliefs toward the COVID-19 vaccine.

*Attitude towards COVID-19 vaccine*. COVID-19 vaccine-related attitudes (α = 0.739) were assessed using six five-point Likert-type items. The total score of attitudes toward the COVID-19 vaccine ranged between 6 and 30, where a higher score indicated higher negative attitudes toward the COVID-19 vaccine.

*Health Belief Model*. The classical HBM consists of the following components: perceived susceptibility, perceived severity, perceived benefits, and perceived barriers.

*Perceived susceptibility*. Two five-point Likert scale questions were used to measure the perceived susceptibility (α = 0.657) of the COVID-19.

*Perceived severity*. The perceived severity of the COVID-19 was measured using two five-point Likert scale questions, which had an α of 0.612.

*Perceived benefits*. Perceived benefits (α = 0.841) of the COVID-19 vaccination were measured using three five-point Likert scale questions.

*Perceived barriers*. The perceived barriers (α = 0.700) of getting the COVID-19 vaccination were measured using six five-point Likert scale questions.

## Statistical analysis

We first employed univariate descriptive statistical analysis [percentage, mean, and standard deviation (SD)]. The Chi-square test and Point-biserial correlation were used to estimate the bivariate level statistics. The statistically significant ($p \leq 0.05$) variables of the bivariate level

were entered into the multiple logistic regression model after checking the assumptions and multicollinearity. The study was designed and reported following strengthening the Reporting of Observational Studies in Epidemiology (STROBE) guidelines [34]. We analyzed the data using the Statistical Product and Service Solutions (SPSS) software, version 26.

## Results

### Characteristics of the participants

The average age of the respondents was 33.7 years, with an SD of 12.9 (**Table 1**). The highest proportion of respondents was from 18 to 24 years (28.9%). Among the respondents, 46.2% were women, while most respondents (86.9%) were Muslims. The married respondents were 61.6%, while only 20.6% had less than a secondary education level. The rural respondents were 64.3%, while 31.9% were from the Dhaka division. More than 30% of the respondents (31.6%) were students and unemployed. The mean household members were 5.0, while the mean household income was 37627 Taka (1 US$ = 84.8 Taka). Table 1 also shows that sample characteristics were almost nationally representative about age, sex, religion, marital status, place of residence, and the mean number of household members (Column 3, Table 1).

### Prevalence of the COVID-19 vaccine hesitancy

Fig 1 shows that 42.9% of the respondents reported that they would surely receive the COVID-19 vaccine, if available for free, while 17.7% would probably receive it. Besides, 22.8% of respondents strongly encouraged their family members to receive the vaccine, while 37.5% encouraged their family members to take the vaccine (**Fig 1**). On the other hand, 12.3% of the respondents stated that they would delay receiving the vaccine, followed by 13.2% who were unsure about what they would do, 7% would probably not receive it, and 6.9% would surely not receive the vaccine. Similarly, if the family or friends were thinking of receiving the COVID-19 vaccine, 16.8% of the respondents supported the statement that they would ask their family members or friends to delay receiving the vaccine. In comparison, 16.9% would not say anything about it, 2.9% would discourage their family members and friends from receiving the vaccine, and 3% would forbid their family members and friends to receive the vaccine.

**Vaccine hesitancy by respondents' background characteristics.** Overall, 46.2% of the respondents had hesitancy to receive the COVID-19 vaccine. The hesitancy was statistically significantly ($p < 0.05$) associated with respondents' religion, education, place of residence, the administrative division of Bangladesh (Table 2). The prevalence of hesitancy was higher among the Muslims, respondents from city corporation areas, and the Khulna division.

**Vaccine hesitancy by behavioral practices to prevent COVID-19.** Fig 2 shows the prevalence of vaccine hesitancy by the participants' behavioral practices towards COVID-19 prevention. It shows that respondents who never wore a mask in going out of home and avoided crowded places had more vaccine hesitancy than their counterparts though these findings were not statistically significant. However, the respondents who were never conscious about using sanitizer or hand wash had significantly higher vaccine hesitancy (45.9%) than those who were regularly conscious (42%).

**Vaccine hesitancy by knowledge about the COVID-19 vaccine and vaccination process.** Fig 3 illustrates the prevalence of vaccine hesitancy by knowledge about the COVID-19 vaccine. The respondents who strongly disagreed that the COVID-19 vaccine has very mild side effects were more hesitant (58.7%) to receive the vaccine than those who agreed (24.9%) with the statement. In addition, the participants who had incorrect knowledge about the vaccination process were more hesitant than those who had correct knowledge (Fig 4). For example,

**Table 1. Background characteristics of the study population (n = 1497).**

| Variables | Study sample, n (%) | National population (%) |
|---|---|---|
| **Age (in years)** | | |
| 18–24 | 432 (28.9) | 20.1 |
| 25–30 | 362 (24.2) | 19.7 |
| 31–39 | 254 (17.0) | 22.7 |
| 40–49 | 236 (15.8) | 18.5 |
| 50+ | 213 (14.2) | 19.1 |
| Mean (SD) | 33.7 (12.9) | |
| **Sex** | | |
| Women | 692 (46.2) | 50.1 |
| Men | 805 (53.8) | 49.9 |
| **Religion** | | |
| Others | 196 (13.1) | 9.3 |
| Muslim | 1301 (86.9) | 90.7 |
| **Marital status** | | |
| Unmarried | 575 (38.4) | 34.8 |
| Married | 922 (61.6) | 65.2 |
| **Education** | | |
| No education | 129 (8.6) | 28.9 |
| Primary | 179 (12.0) | 27.5 |
| Secondary and higher secondary | 448 (29.9) | 43.6 |
| Graduate | 400 (26.7) | |
| Masters and MPhil/PhD | 341 (22.8) | |
| **Place of residence** | | |
| Rural area | 963 (64.3) | 65.0 |
| Urban area (other than city corporation) | 179 (12.0) | 35.0 |
| City Corporation | 355 (23.7) | |
| **Administrative division of Bangladesh** | | |
| Barishal | 114 (7.6) | 5.7 |
| Chattogram | 253 (16.9) | 20.1 |
| Dhaka | 478 (31.9) | 25.1 |
| Khulna | 137 (9.2) | 10.8 |
| Mymensingh | 108 (7.2) | 7.8 |
| Rajshahi | 180 (12.0) | 12.7 |
| Rangpur | 114 (7.6) | 10.9 |
| Sylhet | 113 (7.5) | 6.8 |
| **Occupation** | | |
| Government, private, & NGO sector job | 202 (13.5) | |
| Professional | 277 (18.5) | |
| Homemakers | 348 (23.2) | |
| Students and unemployed | 473 (31.6) | |
| Agriculture and Day Laborer | 102 (6.8) | |
| Others | 95 (6.3) | |
| **Household members,** Mean (SD) | 5.0 (2.0) | 4.6 |
| **Household income,** Mean (SD) | 37627.2 (81295.9) | |

the vaccine hesitancy was higher (48.9%) among the respondents who did not know about the correct doses of the COVID-19 vaccine compared to those who knew the correct doses of the vaccine (38.6%).

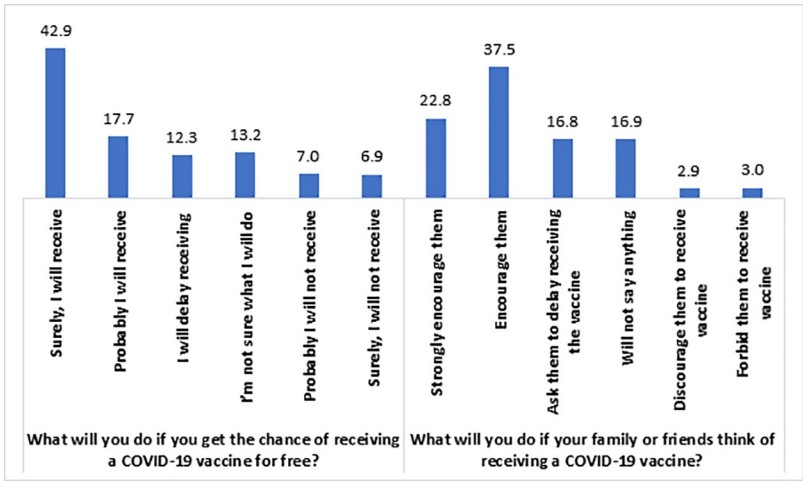

**Fig 1. Prevalence (%) of COVID-19 vaccine hesitancy among the study population (n = 1497).**

**Vaccine hesitancy by attitude towards COVID-19 vaccine and vaccine conspiracy belief.** Table 3 depicts the prevalence of hesitancy by attitudes towards the COVID-19 vaccine and its conspiracy beliefs. The respondents who had more negative attitudes and conspiracy beliefs had more hesitancy to accept the COVID-19 vaccine. For example, the respondents who did not trust the COVID-19 vaccine had more hesitation (59.6%) than those who trusted the COVID-19 vaccination (28.6%). Furthermore, the respondents who believed that the vaccine would probably not work had more hesitancy (62.5%) than those who did not believe it (34.8%). On the other hand, the respondents who strongly agreed that the Coronavirus is a myth to force vaccinations on people had higher hesitancy (61.5%) than those who did not agree with the statement (40.1%). In contrast, the respondents who strongly agreed that the COVID-19 vaccines made in India, America, and Europe are not safer had more hesitancy.

**Vaccine hesitancy by the constructs of health belief model.** The prevalence of vaccine hesitancy by the health belief model components is presented in Table 4. The respondents who strongly disagreed with the statements related to perceived benefits were more hesitant to receive a vaccine. For example, the respondents who strongly disagreed with the statement that "*I think the complications of the COVID-19 will decrease if I get vaccinated and then get infected with the Coronavirus*" had more hesitancy (65.1%) than those who agreed to the statement (31.4%). On the other hand, the respondents who strongly agreed with the statements related to perceived barriers were also more hesitant to receive the COVID-19 vaccine. For instance, the respondents who strongly agreed that registering for the COVID-19 vaccination was difficult for them had more hesitancy (51.8%) than those who disagreed with the statement (38.5%).

## Predictors of COVID-19 vaccine hesitancy

After checking the assumptions and multicollinearity, the significant independent variables at the bivariate level were then entered into the multiple logistic regression model (**Table 5**). We produced three models. The first model included the socio-economic and demographic characteristics of the study population, while the second model included all the variables of model 1 plus knowledge, attitudes, conspiracy beliefs, and behavioral practices related to the COVID-19 vaccine. The third model included all the variables of model 2 plus all the components of HBM. All the regression models were highly significant. The Nagelkerke $R^2$ of the final

**Table 2. COVID-19 vaccine hesitancy by the respondent's characteristics (n = 1497).**

| Variables | Hesitancy (%) | | P-value |
|---|---|---|---|
| | **No** | **Yes** | |
| **Age (in years)** | | | 0.137 |
| 18–24 | 51.6 | 48.4 | |
| 25–30 | 53.0 | 47.0 | |
| 31–39 | 60.6 | 39.4 | |
| 40–49 | 50.4 | 49.6 | |
| 50+ | 55.4 | 44.6 | |
| **Sex** | | | 0.158 |
| Women | 51.9 | 48.1 | |
| Men | 55.5 | 44.5 | |
| **Religion** | | | **<0.001** |
| Others | 68.9 | 31.1 | |
| Muslim | 51.6 | 48.4 | |
| **Marital status** | | | 0.552 |
| Unmarried | 52.9 | 47.1 | |
| Married | 54.4 | 45.6 | |
| **Education** | | | **0.004** |
| No education | 49.6 | 50.4 | |
| Primary | 57.5 | 42.5 | |
| Secondary and higher secondary | 60.5 | 39.5 | |
| Graduate | 49.5 | 50.5 | |
| Masters and MPhil/PhD | 49.9 | 50.1 | |
| **Place of residence** | | | **<0.001** |
| Rural area | 57.5 | 42.5 | |
| Urban area (other than city corporation) | 57.0 | 43.0 | |
| City Corporation | 42.3 | 57.7 | |
| **Administrative division of Bangladesh** | | | **0.004** |
| Barishal | 57.9 | 42.1 | |
| Chattogram | 52.6 | 47.4 | |
| Dhaka | 54.2 | 45.8 | |
| Khulna | 40.1 | 59.9 | |
| Mymensingh | 56.5 | 43.5 | |
| Rajshahi | 59.4 | 40.6 | |
| Rangpur | 46.5 | 53.5 | |
| Sylhet | 63.7 | 36.3 | |
| **Occupation** | | | 0.159 |
| Government, private, & NGO sector job | 48.0 | 52.0 | |
| Professional | 56.3 | 43.7 | |
| Homemakers | 52.3 | 47.7 | |
| Students and unemployed | 53.5 | 46.5 | |
| Agriculture and Day Laborer | 63.7 | 36.3 | |
| Others | 55.8 | 44.2 | |
| **Total** | 53.8 | 46.2 | |

regression model (model 3) was 0.37. Moreover, compared to model 1, successive models had higher $R^2$, and lower -2 Log-likelihood, showing better model fitting.

According to model 3, the Muslims (aOR = 1.80, p ≤ 0.01) and the respondents of city corporation areas (aOR = 2.14, p ≤0.001) were more likely to be hesitant than that of others.

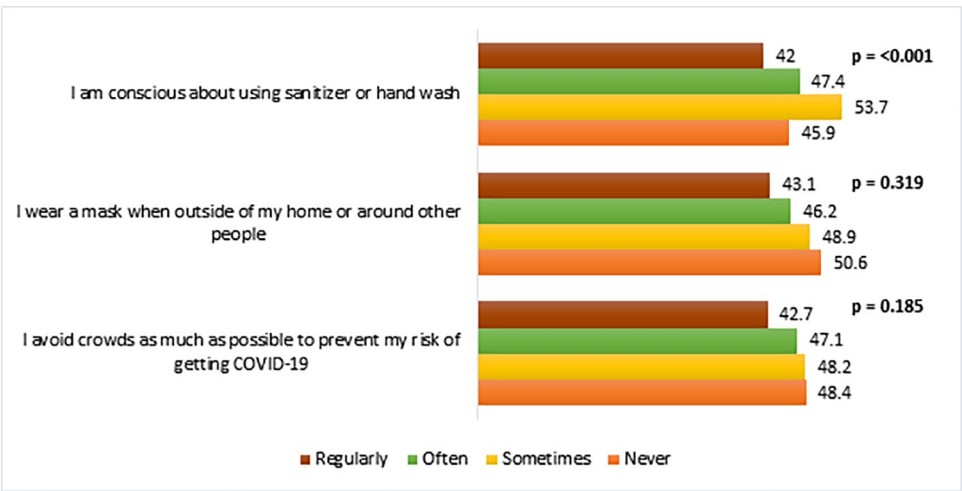

**Fig 2. Vaccine hesitancy (%) by behavioral practices to prevent COVID-19 (n = 1497).**

Compared to the Sylhet division, the participants from Khulna (aOR = 1.31, p ≤0.001) had higher hesitancy. With increasing the knowledge about vaccine (aOR = 0.88, p≤0.001) and knowledge about vaccination process (aOR = 0.91, p ≤ 0.01), hesitancy tended to decrease. On the other hand, with increasing negative attitudes (aOR = 1.17, p ≤0.001) and conspiracy beliefs towards vaccine (aOR = 1.04, p≤0.01), the hesitancy increased. The perceived benefits of COVID-19 vaccination (aOR = 0.85, p ≤0.001) reduced the hesitancy, while perceived barriers (aOR = 1.16, p ≤0.001) increased the hesitancy.

## Discussions

The study found that about 14% of the respondents have asserted their intention not to receive the vaccines, while 16.8% have reported that they would suggest their friends and families delay receiving COVID-19 vaccines. The study also found that 2.9% of the respondents would discourage their family members from receiving the vaccination, and 3% forbid their family members. Overall, this study found a prevalence of 46.2% hesitancy to receive the COVID-19

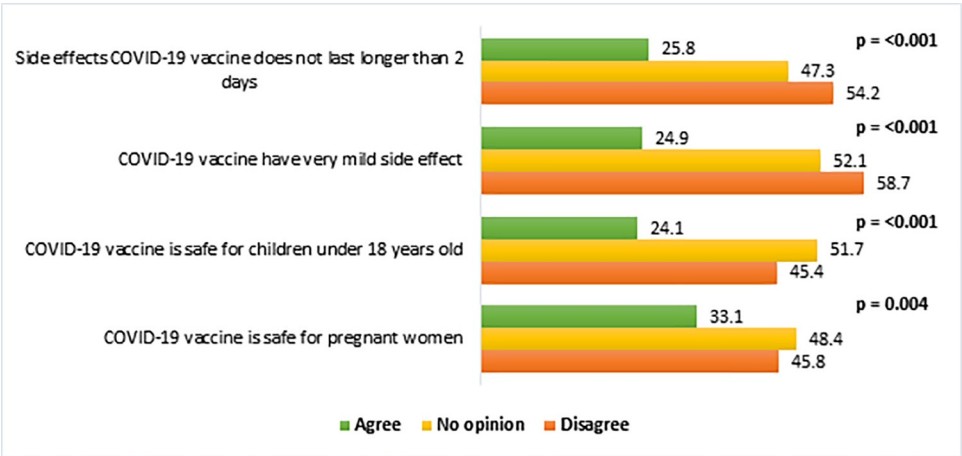

**Fig 3. Vaccine hesitancy (%) by knowledge about the COVID-19 vaccine (n = 1497).**

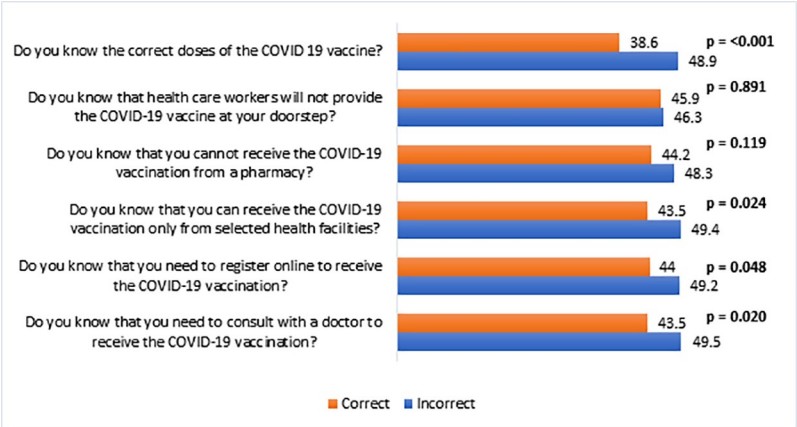

**Fig 4. Vaccine hesitancy (%) by knowledge about the COVID-19 vaccination process (n = 1497).**

vaccine, which is a higher estimate than Kabir et al. (31%) [20], Ali and Hossain (32.5%) [13], and Mahmud et al. (38.8%) [21]. This higher prevalence may partly be explained because the existing studies were conducted in Bangladesh [13, 20, 21] as a rapid assessment of the situation, resulting from participant selection bias. The existing studies also had a small sample size and conducted the online survey [20, 21]. However, data were collected through online and face-to-face interviews from a nationwide sample covering all eight administrative divisions in our study. Therefore, the findings of our study provide a more accurate estimate of COVID-19 vaccine hesitancy among the adult population living in Bangladesh.

**Table 3. Vaccine hesitancy by attitude and conspiracy towards COVID-19 vaccine (n = 1497).**

| Variables and Items | Hesitancy (%) | | | P-value |
|---|---|---|---|---|
| | Disagree [a] | No opinion | Agree [b] | |
| **Attitude towards COVID-19 Vaccine** | | | | |
| I think the COVID-19 vaccine probably will not work | 34.8 | 64.2 | 62.5 | < .001 |
| I do not trust the COVID-19 vaccine | 28.6 | 52.4 | 59.6 | < .001 |
| I think the COVID-19 vaccine is unnecessary | 37.9 | 63.9 | 68.1 | < .001 |
| I think it is not important to get a vaccine to protect people from the COVID-19 | 37.7 | 64.2 | 55.0 | < .001 |
| I do not need a COVID-19 vaccine because I am healthy and at low risk for infection | 30.0 | 59.1 | 62.3 | < .001 |
| I do not need a COVID-19 vaccine because even if I get infected, I will not become seriously ill | 30.8 | 57.6 | 64.6 | < .001 |
| **Conspiracy belief regarding COVID-19 vaccine** | | | | |
| Pharmaceutical companies are encouraging the spread of Coronavirus to make a profit through selling vaccine | 37.0 | 55.8 | 54.4 | < .001 |
| The Coronavirus is a myth to force vaccinations on people | 40.1 | 58.0 | 61.5 | < .001 |
| Drug companies cover up the side effects of vaccines | 29.6 | 53.0 | 60.5 | < .001 |
| People are deceived about the effectiveness of vaccines | 31.4 | 54.1 | 60.7 | < .001 |
| COVID-19 vaccine can result into autism | 33.4 | 52.5 | 52.9 | < .001 |
| A coronavirus vaccination could give one coronavirus | 33.9 | 55.4 | 51.8 | < .001 |
| COVID-19 vaccines made in America and Europe are not safer than those made in other countries | 41.4 | 48.3 | 52.2 | 0.011 |
| COVID-19 vaccines made in China and Russia are not safer than those made in other countries | 40.3 | 48.3 | 45.3 | 0.052 |
| COVID-19 vaccines made in India are not safer than those made in other countries | 26.3 | 46.1 | 53.6 | < .001 |

a. Includes strongly disagree and disagree.

b. Includes strongly agree and agree.

Table 4. Vaccine hesitancy by health belief model related to COVID-19 vaccine (n = 1497).

| Health Belief Model | Hesitancy (%) | | | P-value |
|---|---|---|---|---|
| | Disagree [a] | No opinion | Agree [b] | |
| **Perceived Susceptibility** | | | | |
| I am worried about the likelihood of getting infected by COVID-19 | 50.2 | 57.5 | 38.7 | < .001 |
| I am at high risk of COVID-19 because of my health conditions | 46.7 | 51.1 | 35.5 | 0.001 |
| **Perceived Severity** | | | | |
| I will be very sick if I get infected by COVID-19 | 49.5 | 53.7 | 33.1 | < .001 |
| I am very concerned that I could die from COVID-19 | 47.6 | 48.8 | 38.6 | 0.01 |
| **Perceived Benefits** | | | | |
| I think vaccination is good because it will make me less worried about COVID-19 | 60.2 | 62.8 | 34.9 | < .001 |
| I believe vaccination will decrease my risk of getting infected by COVID-19 | 65.5 | 60.7 | 33.7 | < .001 |
| I think the complications of COVID-19 will decrease if I get vaccinated and then get infected with the Coronavirus. | 65.1 | 56.3 | 31.4 | < .001 |
| **Perceived Barriers** | | | | |
| I am worried that the possible side effects of the COVID-19 vaccination would interfere with my usual activities | 28.2 | 56.3 | 43.6 | < .001 |
| I am concerned about the efficacy of the COVID-19 vaccine | 28.2 | 56.3 | 43.6 | < .001 |
| I have a concern that I may receive faulty/fake COVID-19 vaccine | 27.6 | 47.3 | 51.6 | < .001 |
| It concerns me that the development of a COVID-19 vaccine is too rushed to test its safety properly | 25.8 | 51.7 | 57.9 | < .001 |
| I am concerned about the long-term side effects of the COVID-19 vaccination | 28.4 | 49.2 | 51.1 | < .001 |
| Registering for COVID-19 vaccination is difficult for me | 38.5 | 54.0 | 51.8 | < .001 |

a. Includes strongly disagree and disagree.

b. Includes strongly agree and agree.

Our study shows that religion was significantly associated with vaccine hesitancy, which is in line with other studies of low and middle-income countries from both non-COVID-19 [35] and COVID-19 contexts [36]. The Muslims had more hesitancy about the receipt of coronavirus vaccination in the current study. The notion of considering vaccines as 'medical assault', doubts regarding the ingredients of the vaccines (doubts over the inclusion of ingredients like pork gelatin) may play a role behind the increased hesitancy of Muslim people regarding COVID-19 vaccines [14, 37]. The COVID-19 vaccines have been considered a 'western plot' to sterilize Muslim women in Asian countries like Pakistan. Thus vaccine has been largely discouraged by the community [37, 38]. Similarly, in different earlier non-COVID-19 examples of the middle-income countries like Malaysia, such as in the cases of measles, mumps, and rubella (MMR), religious ruling against vaccines considering them as 'haram' (forbidden) due to the suspected presence of ingredients derived from pigs, receiving vaccines were discouraged [39]. Religious fatalism among the Muslims, including the beliefs that 'everything is in the hands of Allah,' and sense of inability of avoiding death when it is the will of Allah, influences the perception of health among Muslims [40] and such perspectives on health, in this case, is possibly growing vaccine hesitancy among the Muslims [41].

The findings of this study show that respondents from the city corporation areas are more hesitant about the uptake of the COVID-19 vaccines. Due to having more exposure to the different online and offline sources of information, the residents of the city corporation have more possibility of producing fear-driven stigma and conspiracy beliefs regarding COVID-19 vaccines, which may explain their higher level of vaccine hesitancy. In a non-COVID-19 context (dengue vaccine), the broader access towards negative media information in urban areas regarding vaccines has been found responsible for a high level of vaccine hesitancy in other low-middle-income countries, like the Philippines [42]. However, another study on the

**Table 5. Factors affecting COVID-19 vaccine hesitancy among adult population in Bangladesh using multiple logistic regression (n = 1497).**

| Independent variables | Model 1 | Model 2 | Model 3 |
|---|---|---|---|
| | aOR (95% CI) | aOR (95% CI) | aOR (95% CI) |
| **Religion (Others as RC)** | | | |
| Muslim | 2.29 (1.62, 3.22)*** | 2.17 (1.48, 3.18)*** | 1.80 (1.22, 2.66)** |
| **Education (Masters and MPhil/PhD as RC)** | | | |
| No education | 1.41 (0.89, 2.24) | 0.69 (0.40, 1.20) | 0.83 (0.47, 1.46) |
| Primary | 0.97 (0.64, 1.47) | 0.62 (0.38, 1.02) | 0.70 (0.42, 1.17) |
| Secondary and higher secondary | 0.83 (0.60, 1.16) | 0.61 (0.41, 0.90)* | 0.63 (0.42, 0.95) |
| Graduate | 1.24 (0.91, 1.71) | 1.20 (0.84, 1.72) | 1.18 (0.81, 1.70)* |
| **Place of residence (Rural as RC)** | | | |
| Urban area (other than city corporation) | 0.88 (0.62, 1.25) | 1.17 (0.78, 1.75) | 1.19 (0.79, 1.80) |
| City Corporation | 2.06 (1.52, 2.78)*** | 2.00 (1.42, 2.81)*** | 2.14 (1.50, 3.05)*** |
| **Administrative division of Bangladesh (Sylhet as RC)** | | | |
| Barishal | 1.47 (0.92, 2.34) | 1.06 (0.62, 1.82) | 1.19 (0.68, 2.10) |
| Chattogram | 1.05 (0.68, 1.62) | 0.81 (0.49, 1.35) | 0.77 (0.45, 1.29) |
| Dhaka | 2.37 (1.41, 4.00)*** | 1.23 (0.66, 2.27) | 1.31 (0.69, 2.47)*** |
| Khulna | 1.22 (0.71, 2.12) | 0.90 (0.47, 1.74) | 0.88 (0.44, 1.78) |
| Mymensingh | 1.01 (0.61, 1.65) | 0.56 (0.31, 1.01) | 0.59 (0.32, 1.09) |
| Rajshahi | 2.35 (1.35, 4.11) | 1.32 (0.70, 2.49) | 1.32 (0.69, 2.52) |
| Rangpur | 0.84 (0.49, 1.46)** | 0.51 (0.27, 0.97)* | 0.49 (0.25, 0.96) |
| **Behavioral practice to prevent COVID-19** | | 1.00 (0.96, 1.05) | 1.01 (0.96, 1.06) |
| **Knowledge about the COVID-19 vaccine** | | 0.86 (0.81, 0.91)*** | 0.88 (0.82, 0.93)*** |
| **Knowledge about the vaccination process** | | 0.90 (0.84, 0.97) ** | 0.91 (0.84, 0.98) ** |
| **Conspiracy belief regarding COVID-19 vaccine** | | 1.08 (1.05, 1.12) *** | 1.04 (1.01, 1.12) ** |
| **Attitude towards COVID-19 vaccine** | | 1.23 (1.18, 1.27) *** | 1.17 (1.12, 1.22) *** |
| **Health Belief Model** | | | |
| Perceived susceptibility | | | 0.93 (0.85, 1.01) |
| Perceived severity | | | 0.93 (0.85, 1.02) |
| Perceived benefits | | | 0.85 (0.79, 0.91) *** |
| Perceived barriers | | | 1.16 (1.11, 1.22) *** |
| **Model Summary** | | | |
| -2 Log likelihood | 1976.95 | 1649.14 | 1587.36 |
| Cox & Snell R Square | 0.06 | 0.24 | 0.27 |
| Nagelkerke R Square | 0.08 | 0.33 | 0.37 |

aOR: Adjusted Odds Ratio; 95% confidence interval in the parenthesis

* = P≤0.05

** = P≤0.01

*** = P≤0.001; RC = Reference category.

COVID-19 context in Bangladesh found that rural inhabitants were more likely to experience vaccine-related hesitancy than their urban counterparts [22].

Our study shows that the hesitancy decreased with increased knowledge about the COVID-19 vaccine and its associated processes. Thus, the vaccine-related knowledge, which creates awareness regarding vaccine's role in decreasing the risks of the diseases among individuals, plays a role in lessening vaccine hesitancy [43]. Furthermore, being knowledgeable and aware of the vaccine is a significant predictor of vaccine hesitancy in other studies conducted in COVID-19 [44] and non-COVID-19 [45, 46] contexts in lower-middle-income countries like

India and Malaysia. Thus, it leaves ample scopes for circulating evidence-based information about the COVID-19 vaccine among people to increase vaccine uptake.

Attitudes toward the COVID-19 vaccine have been appeared to be one of the strongest predictors of vaccine hesitancy. The negative attitudes towards the COVID-19 vaccine, including perceiving less importance of vaccines, and mistrust about effectiveness, were associated with increasing vaccine hesitancy among the respondents of this study. A negative or anti-COVID-19 vaccination attitude is formed because of the low confidence in vaccine safety [47] and vaccine benefits [48], concerns regarding potential side effects [49], and also the newness of the vaccine [50]. The finding of this current study is in line with other studies conducted in Bangladesh and other lower-middle-income countries, where it was shown that people having a more negative attitude towards the COVID-19 vaccines are less willing to receive the vaccine [22, 51].

The conspiracy beliefs about the COVID-19 vaccines regarding pharmaceutical companies' roles, vaccine manufacturers, and consequences of vaccination have been responsible for increasing the vaccine hesitancy in our study. In various studies conducted in lower-middle-income countries like Pakistan, conspiracy narratives have been regarded as the seed bearer of vaccine hesitancy and considered responsible for resistance against the COVID-19 vaccination programs [14]. Furthermore, the hesitancy towards receiving the COVID-19 vaccines has been significantly influenced by different conspiracy beliefs in some Arab countries like Jordan and Kuwait [15]. Furthermore, various conspiracies, including misinformation regarding the origin of the virus, COVID-19 vaccines trials [15], suspicions around vaccine manufacturers (pharmaceuticals companies and country of origin) [15, 52] regarding vaccine efficacy and safety, have been considered as responsible in other studies conducted in the context of developing countries like Kuwait and Uganda for fueling pre-existing fears, fostering mistrust, doubts, and cynicism over new vaccines, and lowering the COVID-19 vaccination intention of people [53].

The study used the constructs of HBM as independent variables to predict vaccine hesitancy. It was found that perceived benefits and barriers components of HBM were strongly predicting the prevalence of vaccine hesitancy among individuals. Our study found a strong negative association between perceived benefits and the COVID-19 vaccine hesitancy. Considering a particular action (in this case, receiving vaccination) as effective in preventing a disease, which is perceived benefits according to HBM constructs, motivates individuals in adopting the behaviors [54]. On the other hand, perceived barriers were positively associated with vaccine hesitancy in our study. Different perceived structural and attitudinal barriers have been found in other studies conducted in the context of Bangladesh and other developing countries like Egypt [20, 55] as responsible for the vaccine hesitancy, such as lack of information about the vaccination and its adverse effects [55], not getting access to the vaccination coverage [56], affordability issues [57], individual's negative concerns regarding side effects and efficacy of the vaccine [58].

This study explored the prevalence and determinants of the COVID-19 vaccine hesitancy in Bangladesh, which will help the policymakers develop tailored messages to combat the vaccine hesitancy among the people and increase its uptake. However, some limitations of this study should be considered in interpreting the results. First, this study used non-probability sampling; therefore, we should be careful about the generalization of the findings. Second, though this study tried to represent the national population in terms of age, sex, residence, region, marital status, and religion, the distribution of education among the respondents is to some extent not comparable to national data. Third, this study collected self-reported data that may suffer from reporting bias to some extent. Finally, this research used a cross-sectional study design which can not establish causality.

## Conclusions

This nationwide survey provides crucial evidence that nearly half of the adults (46.2%) in Bangladesh are hesitant to receive the COVID-19 vaccine. Thus, the study's findings warrant serious attention of the concerned public health authorities in Bangladesh as the government aims to vaccinate 80% of the total population to bring the pandemic under control. Our findings suggest that negative attitudes, mistrust, and conspiracy beliefs regarding the COVID-19 vaccine are widely prevalent among the people in Bangladesh. Therefore, a vigorous behavior change communication campaign involving community people should be designed and implemented to demystify negative public attitudes towards the vaccine. Besides, it is important to ensure that proper knowledge regarding the COVID-19 vaccine and vaccination process is continuously circulated through effective mass media channels, e.g., online, TV news, and social media. In this regard, public health messaging which emphasis trust in vaccine safety, effectiveness, and benefits can play a significant role. The policymakers should also think about revisiting the policy of the online registration process to receive the COVID-19 vaccine, as we found that online registration is a key structural barrier for many due to the persistent digital divide in the country. It particularly prevents women and the population from the lower socio-economic strata from receiving the COVID-19 vaccine. In this regard, initiatives like text message services using mobile phone operators can ease the registration process as more than 75% of Bangladeshis own a mobile phone. Finally, the government should consider the population's preference regarding vaccines' country of manufacture to reduce vaccine hesitancy and increase voluntary uptake of the COVID-19 vaccine.

## Supporting information

**S1 Map. Study area from where data were collected.**
(TIF)

## Acknowledgments

We would like to thank the participants of this study. We would also like to thank the data collectors for their contribution amidst this challenging time of the COVID-19 pandemic.

## Author Contributions

**Conceptualization:** Mohammad Bellal Hossain, Md. Zakiul Alam, Md. Syful Islam, Shafayat Sultan, Md. Mahir Faysal, Sharmin Rima, Md. Anwer Hossain, Abdullah Al Mamun.

**Data curation:** Mohammad Bellal Hossain, Md. Zakiul Alam, Shafayat Sultan, Md. Mahir Faysal, Sharmin Rima, Md. Anwer Hossain, Abdullah Al Mamun.

**Formal analysis:** Mohammad Bellal Hossain, Md. Zakiul Alam, Shafayat Sultan.

**Investigation:** Mohammad Bellal Hossain, Md. Zakiul Alam, Md. Syful Islam, Shafayat Sultan, Md. Mahir Faysal, Sharmin Rima, Md. Anwer Hossain, Abdullah Al Mamun.

**Methodology:** Mohammad Bellal Hossain, Md. Zakiul Alam, Md. Syful Islam, Shafayat Sultan, Md. Mahir Faysal, Sharmin Rima, Md. Anwer Hossain, Abdullah Al Mamun.

**Project administration:** Mohammad Bellal Hossain, Md. Zakiul Alam, Md. Syful Islam, Shafayat Sultan, Md. Mahir Faysal, Sharmin Rima, Md. Anwer Hossain, Abdullah Al Mamun.

**Resources:** Mohammad Bellal Hossain, Md. Zakiul Alam, Md. Syful Islam, Shafayat Sultan, Md. Mahir Faysal, Sharmin Rima, Md. Anwer Hossain, Abdullah Al Mamun.

**Software:** Md. Zakiul Alam.

**Supervision:** Mohammad Bellal Hossain, Md. Zakiul Alam, Md. Syful Islam, Shafayat Sultan, Md. Mahir Faysal, Sharmin Rima, Md. Anwer Hossain, Abdullah Al Mamun.

**Validation:** Mohammad Bellal Hossain, Md. Zakiul Alam, Shafayat Sultan.

**Visualization:** Mohammad Bellal Hossain, Md. Zakiul Alam, Shafayat Sultan.

**Writing – original draft:** Mohammad Bellal Hossain, Md. Zakiul Alam, Shafayat Sultan.

**Writing – review & editing:** Mohammad Bellal Hossain, Md. Zakiul Alam, Md. Syful Islam, Shafayat Sultan, Md. Mahir Faysal, Sharmin Rima, Md. Anwer Hossain, Abdullah Al Mamun.

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
