## [Decision Letter · Decision Letter 0]

3 Jun 2021

PONE-D-21-14374

COVID-19 Vaccine Hesitancy among the Adult Population in Bangladesh: A Nationally Representative Cross-sectional Survey

PLOS ONE

Dear Dr. Hossain,

Thank you for submitting your manuscript to PLOS ONE. After careful consideration, we feel that it has merit but does not fully meet PLOS ONE’s publication criteria as it currently stands. Therefore, we invite you to submit a revised version of the manuscript that addresses the points raised during the review process.

The manuscript covers an important aspect related to COVID vaccine. I  also suggest you to check for grammatical errors throughout the manuscript

We look forward to receiving your revised manuscript.

Kind regards,

Leeberk Raja Inbaraj, MD

Academic Editor

PLOS ONE

Journal Requirements:

3. PLOS ONE does not copy edit accepted manuscripts (https://journals.plos.org/plosone/s/criteria-for-publication#loc-5). To that effect, please ensure that your submission is free of typos and grammatical errors.

Reviewers' comments:

Reviewer's Responses to Questions

**Comments to the Author**

1. Is the manuscript technically sound, and do the data support the conclusions?

Reviewer #1: Yes

Reviewer #2: Yes

2. Has the statistical analysis been performed appropriately and rigorously? 

Reviewer #1: Yes

Reviewer #2: Yes

3. Have the authors made all data underlying the findings in their manuscript fully available?

Reviewer #1: Yes

Reviewer #2: Yes

4. Is the manuscript presented in an intelligible fashion and written in standard English?

Reviewer #1: Yes

Reviewer #2: Yes

5. Review Comments to the Author

Reviewer #1: The study has been done well. However, some suggestions to improve the paper are as follows.

1. Please mention how the items for the various scales used in the study were developed and validated,

2. Mean score of the hesitancy scale should not be used to calculate hesitancy rather the proportion of participants who were not willing to take/recommend the vaccine to family members should be considered for vaccine hesitancy. Use of the previous method has led to overestimation of vaccine hesitancy in the study.

3. Accuracy of the scale cannot be calculated by dividing the mean score of the scale by overall score (Line 257-259).

4. Comparing the proportion with vaccine hesitancy with the respondent’s characteristics will be more appropriate than mean score of vaccine hesitancy (Table 2). Computing the overall score for each predictor variable and categorising it and comparing it with the proportion with vaccine hesitancy would be more meaningful than comparing mean scores with each question under a predictor variable. Also, the responses of for each question under a predictor variable can be clubbed into categories namely agree, disagree and no opinion and then be compared (Table 3 and Table 4).

Reviewer #2: Comments

At the outset, congratulations to the authors for conducting this timely study that carries public health importance in the context of the ongoing COVID-19 pandemic.

Abstract

- Abstract provides a balanced summary. The methods section under the abstract shall indicate the study setting (total administrative divisions included) and duration of the study.

Methods:

- Describe study setting: total population of Bangladesh, proportion of urban or rural population, sex ratio, life expectancy, literacy rate, economic status, and other factors that have influence on the vaccine hesitancy.

- Please indicate the exact number of participants from the online and face-to-face surveys.

- The sampling strategy used for selecting participants from the districts of administrative divisions of Bangladesh for the face-to-face interviews shall be included. Describe how (sampling procedure) and where (setting) the participants were chosen within each district.

- Please include the reference number and the date of approval of ethics clearance under the ‘ethical approval’ section

- What was the rationale behind choosing margin of error of 0.03 and design effect of 1.6 in sample size calculation?

- Please include what was considered as non-response by an individual during face-to-face interviews?

- Regarding staff involved in face to face interviews: Who were involved in data collection and their training before data collection shall be described in brief.

- The study claims to be a nationally representative survey. However, the methodology described doesn’t reflect a robust sampling strategy to support the claim. The limitations of the study admits the non-probability sampling method used and cautions the generalizability of the study findings. So, kindly justify or changes shall be made as appropriate in the manuscript.

Results:

- The results are described in detail and are in line with the proposed study objectives. However, I would like to see 95% CI for the summary estimates.

- Multiple sub-group analysis was done in this study and I am not sure if the sample size is enough to test for multiple statistical tests. Authors are requested to consult a bio-statisticians for analysis.

- What were the response rates for the online and face-to-face surveys? Kindly include.

- Please add (N=?) in the titles of the tables

Discussion:

- Region specific data (Bangladesh/other LMICs) on determinants of COVID-19 vaccine hesitancy shall be highlighted in the discussion. This could substantiate the study findings as well as broaden the scope of study results in the international context.

- It may be necessary to broaden to the public health dimension i.e. recommendations or possible actions that could be undertaken to deal with the determinants of COVID-19 vaccine hesitancy

6. PLOS authors have the option to publish the peer review history of their article (what does this mean?). If published, this will include your full peer review and any attached files.

Reviewer #1: **Yes: **Sonali Sarkar

Reviewer #2: **Yes: **Sitanshu Sekhar Kar

---

## [Author Response · Author response to Decision Letter 0]

5 Aug 2021

PONE-D-21-14374: COVID-19 Vaccine Hesitancy among the Adult Population in Bangladesh: A Nationwide Cross-sectional Survey

Reviewer 1

Comment 1: Please mention how the items for the various scales used in the study were developed and validated,

Our Response: Thanks for your attention on this important point. We have addressed this issue in the lines between 163-173 of the cleaned version, which reads as follows:

We selected variables and items for constructing scales from the previous studies [12,28–32] and then mixed and customized the different items to develop the scale for the Bangladesh context. The data collection tool was pretested to validate using the face-to-face interview to determine respondents' understanding of the questions, comprehensiveness of the questionnaire, as well as wording, length and sensitivity of the questions. We calculated internal consistency using Cronbach’s alpha (α) to assess the reliability of the items used in the scales. We developed a total of ten scales, of which seven scales had an α between 0.700 to 0.857, and three scales had an α between 0.612 to 0.657. The reliability analysis of seventy percent of our scales was good as the α ranged between 0.7 and 0.8 [33]. The discussion below provides a detailed discussion on scale development, and the items used in these scales are available at https://osf.io/e4xph/. 

Comment 2: Mean score of the hesitancy scale should not be used to calculate hesitancy rather the proportion of participants who were not willing to take/recommend the vaccine to family members should be considered for vaccine hesitancy. Use of the previous method has led to overestimation of vaccine hesitancy in the study.

Our Response: Thanks for this comment. We agree with you and have followed your recommendation and revised the calculation, reflected in the lines between 176-190 of the cleaned version. However, the new calculation increased the hesitancy rate. Now the hesitancy rate is 46.2% which was 41.1% earlier.

Comment 3: Accuracy of the scale cannot be calculated by dividing the mean score of the scale by overall score (Line 257-259).

Our Response: Thanks for this comment, which is also related to the previous comment. We have deleted this line and followed your recommendation as mentioned above.

Comment 4: Comparing the proportion with vaccine hesitancy with the respondent’s characteristics will be more appropriate than mean score of vaccine hesitancy (Table 2). Computing the overall score for each predictor variable and categorising it and comparing it with the proportion with vaccine hesitancy would be more meaningful than comparing mean scores with each question under a predictor variable. Also, the responses of for each question under a predictor variable can be clubbed into categories namely agree, disagree and no opinion and then be compared (Table 3 and Table 4).

Our Response: Thank you for this comment. We have followed you and revised the Table 2, 3, and 4. Accordingly, we have now used logistic regression instead of linear regression. Table 5 shows the findings of logistic regression.

Reviewer 2

Comment 1: Abstract- Abstract provides a balanced summary. The methods section under the abstract shall indicate the study setting (total administrative divisions included) and the duration of the study.

Our Response: Thanks for this comment. We have added this in the abstract. Please see line 31 in the cleaned version.

Comment 2: Methods:- Describe study setting: total population of Bangladesh, proportion of urban or rural population, sex ratio, life expectancy, literacy rate, economic status, and other factors that have influence on the vaccine hesitancy.

Our Response: We appreciate your comment to improve our manuscript by adding this point. We have described the study setting. Please see lines 108 to 119 of the cleaned version.

Comment 3: Please indicate the exact number of participants from the online and face-to-face surveys.

Our Response: Thanks a lot for this important comment. We have added this information. Please see lines 130 to 131 of the cleaned version.

Comment 4: The sampling strategy used for selecting participants from the districts of administrative divisions of Bangladesh for the face-to-face interviews shall be included. Describe how (sampling procedure) and where (setting) the participants were chosen within each district.

Our Response: Thanks a lot for highlighting this point. Please see lines 141 to 160 of the cleaned version where we have addressed your comment.

Comment 5: Please include the reference number and the date of approval of ethics clearance under the ‘ethical approval’ section

Our Response: Please see line 249 of the cleaned version where we have addressed this query.

Comment 6: What was the rationale behind choosing margin of error of 0.03 and design effect of 1.6 in sample size calculation?

Our Response: Thanks a lot for this important question. We have used the prevalence of vaccine hesitancy (p) from a previous study conducted in Bangladesh. The prevalence rate was 32.5% that means the value of p in our formula was 0.325. With this value of p, we could use a margin of error of 0.05, which is normally suggested. However, to increase our sample size considering a nationwide survey, we reduced the margin of error to increase the sample size. On the other hand, the design effect is normally suggested between 1.5 to 3.0. Thus, we adopted 1.6 appropriate for this study. 

Comment 7: Please include what was considered as non-response by an individual during face-to-face interviews?

Our Response: Thanks for this important question. The respondents who did not consent to take part in the study and who consented to participate in the study but did not know about the COVID-19 vaccine were considered as non-response. Please see lines 126 to 130 of the cleaned version where we have addressed this issue.

Comment 8: Regarding staff involved in face to face interviews: Who were involved in data collection and their training before data collection shall be described in brief.

Our Response: Please see lines 157 to 160 of the cleaned version where we have described this.

Comment 9: The study claims to be a nationally representative survey. However, the methodology described doesn’t reflect a robust sampling strategy to support the claim. The limitations of the study admits the non-probability sampling method used and cautions the generalizability of the study findings. So, kindly justify or changes shall be made as appropriate in the manuscript.

Our Response: Thanks a lot for your significant observation. We agree with you and thus, revised our title.

Comment 10: Results:- The results are described in detail and are in line with the proposed study objectives. However, I would like to see 95% CI for the summary estimates.

Our Response: The findings of the adjusted odds ratio have been reported with 95%CI. Please see Table 5.

Comment 11: - Multiple sub-group analysis was done in this study and I am not sure if the sample size is enough to test for multiple statistical tests. Authors are requested to consult a bio-statisticians for analysis.

Our Response: Thanks for your observation. We did not do any sub-group analysis, e.g., different analyses for men and women. We just conducted statistical analysis to establish the association between the outcome variable and independent variables. The analysis was used to select variables for multiple logistic regression. Our team has a Biostatistician, and we are confident that we had enough samples for conducting multiple regression analysis.

Comment 12: What were the response rates for the online and face-to-face surveys? Kindly include.

Our Response: Thanks for this observation. Please see lines 126 to 130 of the cleaned version where we have addressed this issue.

Comment 13: Please add (N=?) in the titles of the tables

Our Response: We have addressed this comment in all the figures and tables. We appreciate your observation.

Comment 14: Discussion:- Region specific data (Bangladesh/other LMICs) on determinants of COVID-19 vaccine hesitancy shall be highlighted in the discussion. This could substantiate the study findings as well as broaden the scope of study results in the international context.

Our Response: Thanks for this advice. Region specific data (Bangladesh/other LMICs) on determinants of COVID-19 vaccine hesitancy is highlighted now in the discussion.

Comment 15: It may be necessary to broaden to the public health dimension i.e. recommendations or possible actions that could be undertaken to deal with the determinants of COVID-19 vaccine hesitancy.

Our Response: Thanks for this significant comment. We have addressed this issue in the conclusion section now.

---

## [Decision Letter · Decision Letter 1]

7 Sep 2021

PONE-D-21-14374R1COVID-19 Vaccine Hesitancy among the Adult Population in Bangladesh: A Nationwide Cross-sectional SurveyPLOS ONE

Dear Dr. Hossain,

Thank you for submitting your manuscript to PLOS ONE. After careful consideration, we feel that it has merit but does not fully meet PLOS ONE’s publication criteria as it currently stands. Therefore, we invite you to submit a revised version of the manuscript that addresses the points raised during the review process.

We look forward to receiving your revised manuscript.

Kind regards,

Leeberk Raja Inbaraj, MD

Academic Editor

PLOS ONE

Journal Requirements:

Reviewers' comments:

Reviewer's Responses to Questions

**Comments to the Author**

1. If the authors have adequately addressed your comments raised in a previous round of review and you feel that this manuscript is now acceptable for publication, you may indicate that here to bypass the “Comments to the Author” section, enter your conflict of interest statement in the “Confidential to Editor” section, and submit your "Accept" recommendation.

Reviewer #1: All comments have been addressed

Reviewer #3: (No Response)

Reviewer #4: All comments have been addressed

2. Is the manuscript technically sound, and do the data support the conclusions?

Reviewer #1: Yes

Reviewer #3: No

Reviewer #4: Yes

3. Has the statistical analysis been performed appropriately and rigorously? 

Reviewer #1: Yes

Reviewer #3: N/A

Reviewer #4: I Don't Know

4. Have the authors made all data underlying the findings in their manuscript fully available?

Reviewer #1: Yes

Reviewer #3: Yes

Reviewer #4: Yes

5. Is the manuscript presented in an intelligible fashion and written in standard English?

Reviewer #1: Yes

Reviewer #3: Yes

Reviewer #4: No

6. Review Comments to the Author

Reviewer #1: Thank you for accepting and modifying the manuscript as per the suggestions. There's a lot of improvement in the manuscript. Though the manuscript has been written in an intelligible way, the language needs corrections at multiple places. The writeup appears to be verbose, can be pruned to convey the same message in lesser words.

Reviewer #3: (No Response)

Reviewer #4: General comments: 1. Sentences need to be framed in correct English, to be understandable by the reader. Line 62-63, Line 94 (should be plural-studies, as we aim to look for multiple related studies to say this fact) not very clear. 2. Table formatting and alignment needed overall. 3. Result section cannot have terms like about, nearly, near to. Be confident about the percentages obtained from your research. 4. Avoid using mass people repeatedly, mass in itself explains it (Line 83-84).

Comment-1 How was sample size decided for online and face to face interview? Needs more clarity on this.

Comment-2 How were participants consented and approached for both the modes of data collection?

Comment-3 Please specify inclusion and exclusion criteria clearly in the beginning of the study methodology.

Study methodology needs more specification regarding study setting-mention names of the blocks in which study was set up. Presently everything is scattered over the places. It will create confusion in readers mind. Have proper sub headings for methodology section.

Comment-4 Fig 3, 4 , Table 3,4 needs percentage inclusion in paragraph written for them, Plain text not making much sense.

Comment-4 Please justify why was non probability sampling was chosen.

7. PLOS authors have the option to publish the peer review history of their article (what does this mean?). If published, this will include your full peer review and any attached files.

Reviewer #1: **Yes: **Sonali Sarkar

Reviewer #3: No

Reviewer #4: **Yes: **rchhokar

---

## [Author Response · Author response to Decision Letter 1]

22 Oct 2021

PONE-D-21-14374R1: COVID-19 Vaccine Hesitancy among the Adult Population in Bangladesh: A Nationwide Cross-sectional Survey

Reviewer 1

Comment 1: Thank you for accepting and modifying the manuscript as per the suggestions. There's a lot of improvement in the manuscript. Though the manuscript has been written in an intelligible way, the language needs corrections at multiple places. The writeup appears to be verbose, can be pruned to convey the same message in lesser words.

Our Response: Thanks a lot for your comments. We have edited the manuscripts according to your suggestion.

Reviewer 4: 

Comment 1: Sentences need to be framed in correct English, to be understandable by the reader. Line 62-63, Line 94 (should be plural-studies, as we aim to look for multiple related studies to say this fact) not very clear. 

Our Response: Thanks a lot, pointing this out. We have taken care of this issue in the manuscript. 

Comment 2: Table formatting and alignment needed overall. 

Our Response: We are sorry that we could not identify any table formatting and alignment issue. We understand that it can be addressed during the production phase if there is any such issue. 

Comment 3: Result section cannot have terms like about, nearly, near to. Be confident about the percentages obtained from your research. 

Our Response: Thanks a lot for raising this critical point. We are confident about the percentages that we have reported in the table. We wrote in that style to avoid numbers at the beginning of the sentence. However, we have revised the manuscript thoroughly to address your comment. 

Comment 4: Avoid using mass people repeatedly, mass in itself explains it (Line 83-84).

Our Response: Thank you. We have addressed this.

Comment 5: How was sample size decided for online and face to face interview? Needs more clarity on this.

Our Response: We decided to follow a 2:1 ratio for interviewing samples using face-to-face and the online survey considering the country's digital divide. We have provided this justification. 

Comment 6: How were participants consented and approached for both the modes of data collection?

Our Response: Thanks a lot for raising this question. We have added a description of this issue in the Ethical Approval Participant’s Consent section.

Comment 7: Please specify inclusion and exclusion criteria clearly in the beginning of the study methodology. Study methodology needs more specification regarding study setting-mention names of the blocks in which study was set up. Presently everything is scattered over the places. It will create confusion in readers mind. Have proper sub headings for methodology section.

Our Response: We have thoroughly worked on these issues and revised our methodology section.

Comment 8: Fig 3, 4 , Table 3,4 needs percentage inclusion in paragraph written for them, Plain text not making much sense.

Our Response: Thanks a lot for this suggestion. We have revised figures 2 to 5 and tables 3 and 4.

Comment 9: Please justify why was non probability sampling was chosen.

Our Response: Thanks a lot for this question. Now, we have justified the adoption of non-probability sampling.

---

## [Decision Letter · Decision Letter 2]

15 Nov 2021

PONE-D-21-14374R2COVID-19 Vaccine Hesitancy among the Adult Population in Bangladesh: A Nationwide Cross-sectional SurveyPLOS ONE

Dear Dr. Hossain,

Thank you for submitting your manuscript to PLOS ONE. After careful consideration, we feel that it has merit but does not fully meet PLOS ONE’s publication criteria as it currently stands. Therefore, we invite you to submit a revised version of the manuscript that addresses the points raised during the review process.

We look forward to receiving your revised manuscript.

Kind regards,

Leeberk Raja Inbaraj, MD

Academic Editor

PLOS ONE

Journal Requirements:

Additional Editor Comments (if provided):

Thank you for revising the manuscript patiently. The manuscript has a good potential for acceptance. However, we would like you to focus on the language and address the the minor comments by the reviewers. I look forward for your revised version.

Reviewers' comments:

Reviewer's Responses to Questions

**Comments to the Author**

1. If the authors have adequately addressed your comments raised in a previous round of review and you feel that this manuscript is now acceptable for publication, you may indicate that here to bypass the “Comments to the Author” section, enter your conflict of interest statement in the “Confidential to Editor” section, and submit your "Accept" recommendation.

Reviewer #1: All comments have been addressed

Reviewer #4: All comments have been addressed

2. Is the manuscript technically sound, and do the data support the conclusions?

Reviewer #1: Yes

Reviewer #4: Yes

3. Has the statistical analysis been performed appropriately and rigorously? 

Reviewer #1: Yes

Reviewer #4: I Don't Know

4. Have the authors made all data underlying the findings in their manuscript fully available?

Reviewer #1: Yes

Reviewer #4: Yes

5. Is the manuscript presented in an intelligible fashion and written in standard English?

Reviewer #1: Yes

Reviewer #4: Yes

6. Review Comments to the Author

Reviewer #1: (No Response)

Reviewer #4: Thank you for modifying the manuscript, it looks much better now.

Comment-1: Line 177-it should be respondents, remove single quotation mark

Comment 2: line 178-179, 181,183- double quotation marks instead of single are recommended

comment 3: line 203-204, 209, single quotation can be removes, as in other places in the same paragraph.

Please check for similar changes across the manuscript if present.

Comment 4: table -1, "Total" in the end is not well clear. In my opinion it can be removed, as table heading has n=1497 mentioned already

Comment 5: table 2 ,3 and 4, in the table title "%", can be removed, as the table below shows data well in % as mentioned in the column 2 of the table. Again, "Total ", not required in the end of the table in my opinion.

Comment 6: Line 364-365, "I think the complications.....Coronavirus", should be in quotations.

Comment 7:Footnote of the table 4 can be without quotation

Comment 8:Line 88 mentions HBM, need not repeat the full form in Line 246.

7. PLOS authors have the option to publish the peer review history of their article (what does this mean?). If published, this will include your full peer review and any attached files.

Reviewer #1: No

Reviewer #4: No

---

## [Author Response · Author response to Decision Letter 2]

15 Nov 2021

PONE-D-21-14374R2: COVID-19 Vaccine Hesitancy among the Adult Population in Bangladesh: A Nationwide Cross-sectional Survey

Reviewer 4: 

Comment-1: Line 177-it should be respondents, remove single quotation mark.

Our Response: Thanks a lot for identifying this typo. We have removed the single quote.

Comment 2: Line 178-179, 181,183- double quotation marks instead of single are recommended.

Our Response: We appreciate your observation and we have modified accordingly. 

Comment 3: Line 203-204, 209, single quotation can be removes, as in other places in the same paragraph. Please check for similar changes across the manuscript if present.

Our Response: We accept your observations and revised the manuscript accordingly. Thanks a lot.

Comment 4: Table -1, "Total" in the end is not well clear. In my opinion it can be removed, as table heading has n=1497 mentioned already.

Our Response: We agree with you and deleted the total. 

Comment 5: Table 2 ,3 and 4, in the table title "%", can be removed, as the table below shows data well in % as mentioned in the column 2 of the table. Again, "Total ", not required in the end of the table in my opinion.

Our Response: Thanks a lot for your comments. We agree about the deletion of % from the table title of tables 2, 3, and 4. However, we believe the “Total” in table 2 is required as the reader will not be able to get this information from the rest of the table. 

Comment 6: Line 364-365, "I think the complications.....Coronavirus", should be in quotations.

Our Response: Thanks a lot! We have used the double quote mark. 

Comment 7: Footnote of the table 4 can be without quotation.

Our Response: Thanks a lot! We have removed the quotation from the Footote. 

Comment 8: Line 88 mentions HBM, need not repeat the full form in Line 246.

Our Response: Thanks a lot! We have corrected this.

---

## [Editor Report · Decision Letter 3]

18 Nov 2021

COVID-19 Vaccine Hesitancy among the Adult Population in Bangladesh: A Nationwide Cross-sectional Survey

PONE-D-21-14374R3

Dear Dr. Hossain,

We’re pleased to inform you that your manuscript has been judged scientifically suitable for publication and will be formally accepted for publication once it meets all outstanding technical requirements.

Kind regards,

Leeberk Raja Inbaraj, MD

Academic Editor

PLOS ONE
---

## [Editor Report · Acceptance letter]

1 Dec 2021

PONE-D-21-14374R3 

COVID-19 Vaccine Hesitancy among the Adult Population in Bangladesh: A Nationwide Cross-sectional Survey 

Dear Dr. Hossain:

I'm pleased to inform you that your manuscript has been deemed suitable for publication in PLOS ONE. Congratulations! Your manuscript is now with our production department. 

Kind regards, 

on behalf of

Dr. Leeberk Raja Inbaraj 

Academic Editor

PLOS ONE